# The Effect of Alkyl Substituents on the Formation and Structure of Homochiral (*R**,*R**)-[R_2_Ga(*µ*-OCH(Me)CO_2_R′)]_2_ Species—Towards the Factors Controlling the Stereoselectivity of Dialkylgallium Alkoxides in the Ring-Opening Polymerization of *rac*-Lactide

**DOI:** 10.3390/molecules30010190

**Published:** 2025-01-06

**Authors:** Magdalena Kaźmierczak, Łukasz Dobrzycki, Maciej Dranka, Paweł Horeglad

**Affiliations:** 1Faculty of Chemistry, Warsaw University of Technology, Noakowskiego 3, 00-664 Warsaw, Poland; kazmierczak.magdalena.m@gmail.com (M.K.); maciej.dranka@pw.edu.pl (M.D.); 2Centre of New Technologies, University of Warsaw, Banacha 2c, 02-097 Warsaw, Poland; 3Faculty of Chemistry, University of Warsaw, Pasteura 1, 02-093 Warsaw, Poland; lm.dobrzycki@gmail.com

**Keywords:** biodegradable polyesters, polylactide, tacticity, heterotactic, dimers, diastereomers

## Abstract

Building on our previous studies, which have demonstrated that homochiral propagating species—(*R**,*R**)-[Me_2_Ga(*µ*-OCH(Me)CO_2_R)]_2_—were crucial for the heteroselectivity of [Me_2_Ga(*µ*-OCH(Me)CO_2_Me)]_2_ in the ring-opening polymerization (ROP) of racemic lactide (*rac*-LA), we have investigated the effect of alkyl groups on the structure and catalytic properties of dialkylgallium alkoxides in the ROP of *rac*-LA. Therefore, we have isolated and characterized the *rac*-[R_2_Ga(*µ*-OCH(Me)CO_2_Me]_2_ (R = Et (**1**), *^i^*Pr (**2**) and *rac*-[R_2_Ga(*µ*-OCH(Me)C_5_H_4_N]_2_ (R = Et (**3**), *^i^*Pr (**4**)) complexes, to demonstrate the effect of alkyl groups on the chiral recognition induced the formation of the respective homochiaral species—(*R**,*R**)-[R_2_Ga(*µ*-OCH(Me)CO_2_Me)]_2_ and (*R**,*R**)-[R_2_Ga(*µ*-OCH(Me)C_5_H_4_N]_2_. Moreover, we have investigated the structure of (*S*,*S*)-[R_2_Ga(*µ*-OCH(Me)CO_2_Me]_2_ (R = Et ((*S*,*S*)-**1**, R = *^i^*Pr ((*S*,*S*)-**2**,) and their catalytic activity in the ROP of *rac*-LA. With an increase in the bulkiness of alkyl substituents on gallium the following can be observed: (a) the tendency for the formation of homochiral complexes decreased, (b) the symmetry of homochiral (*S*,*S*)-[R_2_Ga(*µ*-OCH(Me)CO_2_Me]_2_ (M = Me, Et (*S*,*S*)-**1**), *^i^*Pr (*S*,*S*)-**2**) changed, and both have resulted in (c) lower or no heteroselectivtity across these complexes in the ROP of *rac*-LA. Importantly, the results have confirmed the crucial role of the chiral-induced formation of homochiral asymmetric dimers on the heteroselectivity of dialkylgallium alkoxides in the ROP of *rac*-LA.

## 1. Introduction

Over the last two decades, there has been great interest in stereoselective catalysts/catalytic systems for the ring-opening polymerization (ROP) of *rac*-lactide (*rac*-LA) [1,2,3,4], in order to synthesize PLA of different stereostructures and properties [5,6]. Many efforts have been made in order to determine the effect of the various elements of the structure of metal alkoxide propagating species in the ROP of *rac*-LA on their stereoselectivity, resulting in the development of numerous metal alkoxides capable of the isoselective or heteroselective ROP of *rac*-LA [2,7,8,9]. Notably, there have been considerably fewer reports concerning dimeric metal alkoxides, which can polymerize *rac*-LA in a stereoselective fashion [10,11,12], leading to isotactically [13,14,15,16] and heterotactically [17,18,19,20,21,22,23] enriched PLA in comparison with monomeric species [2,7,8,9]. Irrespective of the dimeric/monomeric structure of the stereoselective propagating species, there are two widely accepted mechanisms of stereoselectivity control in the coordination-insertion polymerization of *rac*-lactide, as well as other chiral and prochiral monomers, i.e., enantiomorphic site control (ESM) and chain end control (CEM) mechanisms [24,25]. They define two main structural elements responsible for the stereoselectivity of both monomeric and dimeric propagating species—the presence of asymmetry on the metal center and the presence of chiral carbon on the last inserted monomeric unit in the growing PLA chain. However, the cooperativity between metal centers, which can be observed only in the case of dimeric catalysts (or higher aggregates), and which has been recently reported for the dimeric metal alkoxides in the ROP of *rac*-LA [12,26,27,28], should be considered an important factor affecting their stereoselectivity. We have demonstrated that the cooperative effects between two metal centers of dimeric gallium and indium alkoxides manifested in the change in the stereoselectivity of the investigated complexes in the ROP of *rac*-LA [1]. In this case, the formation of excess homochiral dimeric species was responsible for the heteroselectivity of dimethylgallium and dimethylindium alkoxides in the ROP of *rac*-lactide, with a positive non-linear effect. With regard to the latter, Nozaki and coworkers have shown that excess homochiral zinc dimeric complexes were responsible for the stereoselective copolymerization of olefines with CO_2_ [29]. Therefore, one should consider the formation of the excess of homochiral dimeric species over heterochiral ones in the ROP, a strategy for stereoselectivity control that is comparable to enantioselective organic transformations, where excess *R*,*R* or *S*,*S* dimeric catalytic species are decisive for the enantiomeric excess (*ee*) of the product [30]. Interestingly, the difference in the stereoselectivity between homochiral and heterochiral dimeric metal alkoxide was reported in the case of Y [31], Cu [32], and In [33,34] complexes. Moreover, the formation of homochiral dimeric Zr and Hf [35], Sc [36], and In [37] complexes could have been responsible for their stereoselectivity, although it was not clearly indicated. Notably, the different activities of homochiral and heterochiral dimeric aluminum alkoxides have been recently observed in the ROP of cyclic esters [38]. Therefore, the factors influencing the formation of homochiral/heterochiral dimeric metal alkoxides should be considered an interesting area of research.

With regard to our previous studies, we have considered dialkylgallium alkoxides, which tend to form dimers both in the solid state and solution [1,39], a good starting point for further research on factors controlling the formation of homochiral/heterochiral dimers, and subsequently, their catalytic activity in the ROP of *rac*-LA. Although the selected class of complexes has been thoroughly investigated with regard to their synthesis and structure [39], the issue concerned with the formation of homochiral/heterochiral species has been barely discussed, not only for dialkylgallium derivatives, but also for group 13 metal dialkylalkoxides. Except for our studies on [Me_2_M(*µ*-CH(Me)CO_2_Me]_2_ (M = Ga, In) with chiral centers on an alkoxide ligand [1], the chiral recognition leading to exclusively homochiral dimers was reported in the case of the analogous dimethylaluminum derivatives of *rac*-ethyl lactate [40]. However, although *rac*-[Me_2_Al(*µ*-CH(Me)CO_2_Et]_2_ mimicked propagating species in the ROP of *rac*-LA with dialkylaluminum alkoxides, they did not allow for the stereoselective polymerization of *rac*-LA due to an essential lack of activity at mild conditions and extensive transesterification observed at higher temperatures [41]. Therefore, with regard to our studies on the structure and catalytic activity of dialkylgallium and dialkylindium alkoxides in the ROP of *rac*-LA [1,42], we have focused on the effect of the alkyl groups of [R_2_Ga(*µ*-CH(Me)CO_2_Me]_2_, on the formation and structure of homochiral species, and their impact on their stereoselectivity in the ROP of *rac*-LA. Studies on the dialkylgallium alkoxides should be particularly interesting, given the high stereoselectivity in the ROP of *rac*-LA and the low tendency for transesterification side reactions in comparison with aluminum [41] and indium [1] analogues.

Hereby, we report the synthesis, structure, and catalytic activity of *rac*-[R_2_Ga(*µ*-OCH(Me)CO_2_Me)]_2_ and (*S*,*S*)-[R_2_Ga(*µ*-OCH(Me)CO_2_Me)]_2_ (R = Et, *^i^*Pr) in the ROP of *rac*-LA, as well as the synthesis and structure of *rac*-[R_2_Ga(*µ*-OCH(Me)C_5_H_4_N]_2_ (R = Et, *^i^*Pr). We discuss how the effect of alkyl substituents affects their structure, most importantly with regard to the tendency for the formation of homochiral complexes. Based on structure-activity studies, we demonstrate that the decrease in the tendency for the formation of homochiral (*R**,*R**)-[R_2_Ga(*µ*-OCH(Me)CO_2_PLA)]_2_ (PLA—growing polylactide chain) considerably influences the stereoselectivity of dialkylgallium alkoxide species in the ROP of *rac*-LA. This constitutes further evidence for the crucial role of homochiral dimeric dialkylgallium alkoxide species in the stereoselective ROP of *rac*-lactide.

## 2. Results and Discussion

In order to demonstrate how gallium-bonded alkyl groups affect the structure and catalytic properties of dialkylgallium alkoxide propagating species in the ROP of *rac*-lactide (*rac*-LA), we initially focused on the synthesis and structure of (*S*,*S*)-[R_2_Ga(*µ*-OCH(Me)CO_2_Me]_2_ and *rac*-[R_2_Ga(*µ*-OCH(Me)CO_2_Me]_2_ (R = Et, *^i^*Pr), as well as *rac*-[R_2_Ga(*µ*-OCH(Me)C_5_H_4_N]_2_ (R = Et, *^i^*Pr), with regard to their dimethylgallium analogues, previously reported by us [1]. Due to the presence of a methyl lactate ligand (*melac*), which is expected to mimic the growing PLA chain [41,43,44,45], both *rac*-[R_2_Ga(*µ*-OCH(Me)CO_2_Me]_2_ (R = Et (**1**), *^i^*Pr (**2**) and (*S*,*S*)-[R_2_Ga(*µ*-OCH(Me)CO_2_Me]_2_ (R = Et ((*S*,*S*)-**1**, R = *^i^*Pr ((*S*,*S*)-**2**,) are expected to represent model complexes for the dialkylgallium propagating species in the ROP of *rac*-LA (Figure 1). Therefore, the studies of (*S*,*S*)-**1** and (*S*,*S*)-**2** were expected to show the effect of alkyl groups on the structure of homochiral—(*R**,*R**)-[R_2_Ga(*µ*-OCH(Me)CO_2_PLA)]_2_—species, which were decisive for the stereoselective ROP of *rac*-LA with analogous dimethylgallium alkoxides [1]. On the other hand, the structures of *rac*-[R_2_Ga(*µ*-OCH(Me)CO_2_Me]_2_ (R = Et (**1**), *^i^*Pr (**2**)) were expected to demonstrate the effect of alkyl groups on the ratio of homochiral—(*R**,*R**)-[R_2_Ga(*µ*-OCH(Me)CO_2_PLA)]_2_—to heterochiral—(*R*,*S*)-[R_2_Ga(*µ*-OCH(Me)CO_2_PLA)]_2_ (R = Et, *^i^*Pr)—propagating species in the ROP of *rac*-LA, both in the absence and presence of pyridine derivatives [1]. The latter was previously used to enhance the formation of the homochiral (*R**,*R**)-[Me_2_Ga(*µ*-OCH(Me)CO_2_PLA)]_2_ species. Finally, *rac*-[R_2_Ga(*µ*-OCH(Me)C_5_H_4_N]_2_ (R = Et (**3**), *^i^*Pr (**4**)), where OCH(Me)C_5_H_4_N represents an alkoxide ligand with pyridine functionality, further served as model complexes in order to demonstrate the effect of alkyl groups on the formation of homochiral/heterochiral dialkylgallium alkoxides in the presence of pyridine derivatives. In addition to the main scope of our research presented above, it should be noted that there is a paucity of data on diethylgallium alkoxides [46,47] or aryloxides [48,49,50,51,52,53,54], and, as far as we are aware, there are no fully characterized di-*iso*-propylgallium alkoxides/aryloxides, in contrast to dimethylgallium or even bulky di-*tert*-butylgallium analogues [39].

*rac*-[R_2_Ga(*µ*-OCH(Me)CO_2_Me]_2_ (R = Et (**1**) and *^i^*Pr (**2**)) and (*S*,*S*)-[R_2_Ga(*µ*-OCH(Me)CO_2_Me)]_2_ (R = Et ((*S*,*S*)-**1**), *^i^*Pr (*S*,*S*-**2**)) complexes were synthesized in the equimolar reaction between R_3_Ga (R = Et, *^i^*Pr) and *rac*-methyl lactate or *S*-methyl lactate, respectively (Figure 2). *rac*-[R_2_Ga(µ-OCH(Me)C_6_H_4_N]_2_ (R = Et (**3**) and *^i^*Pr (**4**)) were synthesized analogously, in the reaction of trialkylgallium compounds with *rac*-HOCH(Me)C_6_H_4_N (Figure 3). All complexes were purified by crystallization and isolated as colorless crystals in moderate to high yields. Single crystals suitable for X-ray analysis were obtained for complexes **2**, **3**, **4**, as well as (*S*,*S*)-**1** and (*S*,*S*)-**2**.

### 2.1. Structure of rac-[R_2_Ga(µ-OCH(Me)CO_2_Me]_2_ (R = Et (***1***), ^i^Pr (***2***))

*rac*-[*^i^*Pr_2_Ga(*µ*-OCH(Me)CO_2_Me)] (**2**) crystallized as centrosymmetric dimer (*R*,*S*)-**2** (Figure 1), similarly to the previously reported (*R*,*S*)-[Me_2_Ga(*µ*-OCH(Me)CO_2_Me)] complex [1]. The coordination sphere of gallium atoms adopted a distorted trigonal-bipyramidal geometry, with the methine carbons of *^i^*Pr groups and alkoxide oxygen of the *melac* ligand defining the equatorial plane. The carbonyl oxygen of *melac* ester functionality and the bridging alkoxide oxygen of the second monomeric unit were located in the axial positions, which is typical for five-coordinate group 13 dialkylalkoxides [39], analogous to dialkylaluminum complexes [55]. The presence of *^i^*Pr resulted in the significantly longer Ga–C bonds (1.9837(15) Å and 1.9899(15) Å), in comparison to Ga–C bonds (1.9584(18) Å and 1.9611(17) Å) of the analogous dimethyl derivative—(*R*,*S*)-[Me_2_Ga(*µ*-OCH(Me)CO_2_Me)] [1]. While this observation is in line with the structure of dialklylgallium alkoxides bearing ethyl and *tert*-butyl alkyl groups [39], it should not be surprising as it can be associated with both larger steric hindrances of *^i^*Pr groups, and especially with the lower basicity of monoanionic (CH_3_)_2_HC^−^ ligands in comparison with H_3_C^−^. Notably, the increase in the steric hindrances of a gallium-bonded alkyl groups did not result in the longer Ga…O(2) chelate interaction (2.449(5) Å) between the gallium and carbonyl group of *melac* ligand in comparison with the analogous dimethyl derivative 2.5220(13) Å. Moreover, the stronger chelate bond in the case of the di-*iso*-propyl derivative **2** could be indicated by the distance of bridging Ga–O bonds in the trans position to chelate Ga–O=C ones [55]. In the latter case, the longer bridging Ga–O bond (2.0547(10) Å) in the *trans* position to the chelate Ga–O(2) bond (2.449(5) Å) was observed in comparison with the analogous distances of 2.0231(11) Å within the Ga_2_O_2_ central ring of (*R*,*S*)-[Me_2_Ga(*µ*-OCH(Me)CO_2_Me)] [1]. Notably, with regard to the discussion on the structure of (*S*,*S*)-[R_2_Ga(*µ*-OCH(Me)CO_2_Me)] (R = Me, Et ((*S*,*S*)-**1**), *^i^*Pr ((*S*,*S*)-**2**))) (see below), the presence of *iso*propyl groups in (*R*,*S*)-**2** had a considerable effect on the Ga(1)–O(1)–Ga(1^1^)–C(2)/Ga(1^1^)–O(1^1^)–Ga(1)–C(2^1^) torsion angles (164.94(14)°/−164.94(14)° in comparison with the analogous angles of (*R*,*S*)-[Me_2_Ga(*µ*-OCH(Me)CO_2_Me)] (146.08(15)°/−146.69(15)°).

With regard to our studies on the structure of [Me_2_Ga(*µ*-OCH(Me)CO_2_Me)]_2_ in solution [1], ^1^H NMR spectra of **1** and **2** were expected to demonstrate the effect of alkyl groups on the distribution of homochiral—(*R**,*R**)-[R_2_Ga(*µ*-OCH(Me)CO_2_Me)]_2_ (R = Et, *^i^*Pr)—and heterochiral—(*R*,*S*)-[R_2_Ga(*µ*-OCH(Me)CO_2_Me)]_2_ (R = Et, *^i^*Pr)—species, as they have been crucial for the heteroselectivity of dialkylgallium alkoxide species in the ROP of *rac*-LA. On the other hand, the FTIR studies should be considered as indicative of the strength of the Ga…O=C chelate interaction between gallium and the *melac* ligand. Notably, in the latter case, the FTIR spectra of (*S*,*S*)-**2** and (*S*,*S*)-**1**, which are discussed below, showed essentially no effect across gallium-bonded ethyl or *iso*-propyl groups in comparison with (*S*,*S*)-[Me_2_Ga(*µ*-OCH(Me)CO_2_Me)]_2_ [42]. The dissolution of (*R*,*S*)-**2** led to the equimolar mixture of (*R**,*R**)-**2** and (*R*,*S*)-**2** dimers, which was revealed by ^1^H NMR and the presence of two sets of signals that are characteristic of homochiral and heterochiral species. The presence of three septets at 1.01, 1.08, and 1.17 ppm, in a 1:2:1 ratio, corresponding to methine protons of Ga–CH(CH_3_)_2_ groups, was especially indicative of their ratio. Analogously, the equimolar mixture of (*R**,*R**)-**1** and (*R*,*S*)-**1**, in the solution of **1** was evidenced by two quartets at 0.61 and 0.77 ppm, and a multiplet at 0.69 ppm, in a 1:1:2 ratio, corresponding to methylene protons of the Ga-CH_2_CH_3_ groups of the respective (*R*,*S*)-**1** and (*R**,*R**)-**1** species. In the case of both **1** and **2**, it confirmed the presence of dimeric species, while the observed equimolar mixture of homochiral and heterochiral species was analogous to *rac*-[Me_2_Ga(*µ*-OCH(Me)CO_2_Me]_2_ [1]. Similarly to the latter, the fast exchange between (*R*,*S*)-[R_2_Ga(*µ*-OCH(Me)CO_2_Me]_2_ and (*R**,*R**)-[R_2_Ga(*µ*-OCH(Me)CO_2_Me]_2_ species was expected. Notable, in the case of **2**, the minor septet, overlapping with the septet at 0.77 ppm and corresponding to (*R*,*S*)-**2** (Figure 2), could suggest the presence of monomeric *^i^*Pr_2_Ga(*µ*-OCH(Me)CO_2_Me) (<4% based on the integration). In this case, the increased tendency of **2** to dissociate in solution was in line with the significantly longer bridging Ga–O′ bond in the Ga_2_O_2_ ring in comparison with the analogous Ga–O bridging bond of *rac*-[Me_2_Ga(*µ*-OCH(Me)CO_2_Me)]_2_ [1]. However, even in the case of **2**, with the bulkiest alkyl groups among the discussed [R_2_Ga(*µ*-OCH(Me)CO_2_Me)] (R = Me, Et, *^i^*Pr), dimeric species were dominant, which also the case in the presence of different pyridine derivatives (Figure 2).

In the case of **1**, the ratio of (*R**,*R**)-**1**/(*R*,*S*)-**1** increased with the addition of differently substituted pyridines, i.e., pyridine, 4-methylpyridine, and 4-(dimethylamino)pyridine, and this was strongly dependent on both their donor number [56] and pyridine/**1** ratio (Figure 3). Notably, the formation of only traces of tentatively assumed monomeric Ga species—Et_2_Ga(OCH(Me)CO_2_Me)(*Py*)—facilitated by the presence of pyridine derivative, could be suggested by the ^1^H NMR spectra (Appendix A.) Importantly, the excess of homochiral (*R**,*R**)-[Et_2_Ga(*µ*-OCH(Me)CO_2_Me)]_2_ ((*R**,*R**)-**1**) species was much lower in comparison with analogous (*R**,*R**)-[Me_2_Ga(*µ*-OCH(Me)CO_2_Me)]_2_ [1], as previously reported for *rac*-[Me_2_Ga(*µ*-OCH(Me)CO_2_Me)]_2_/*Py* (*Py*—pyridine derivative), (Figure 3). On the other hand, no excess homochiral (*R**,*R**)-**2**, was observed under the same conditions, irrespective of the donor number and excess of pyridine derivative added to **2** (Figure 2). On the contrary, the slight excess of heterochiral ((*R*,*S*)-[R_2_Ga(*µ*-OCH(Me)CO_2_Me)]_2_ was observed either without or with, e.g., the pyridine derivative added (Figure 2), which resulted in the (*R**,*R**)-**2/**(*R*,*S*)-**2** ratio of 0.45—0.48, essentially, irrespective of the donor number of the pyridine derivative or *Py*:**2** ratio. Interestingly, the percentage of tentatively assumed monomeric Ga species increased with both the donor number of pyridine or 4-methylpyridine and the Lewis base/**2** ratio, reaching up to 17% in the case of the 4-methylpyridine/**2** ratio of 60:1. In the case of DMAP/**2**, a ratio of 6:1 was observed, where DMAP represents a 4-(dimethylamino)pyridine of the highest donor number among all pyridine derivatives used, and the percentage of monomeric gallium species reached over 75%. The formation of such tentatively proposed monomeric species could be facilitated by the formation of *^i^*Pr_2_Ga(OCH(Me)CO_2_Me)(*Py*), where *Py* represents the pyridine derivative. Although no monomeric dialkylgallium alkoxides could be isolated in our studies, several X-ray structures of *^t^*Bu_2_GaOPh(*Py*) have been reported and well characterized so far [57]. With regard to the latter, further studies on the synthesis and structure of monomeric *^i^*Pr_2_Ga(OCH(Me)CO_2_Me)(*Py*) should be conducted. However, the dimeric species decisive for the stereoselectivity of dialkylgallium alkoxides are the primary focus of this article. While in the case of *rac*-[R_2_Ga(*µ*-OCH(Me)CO_2_Me)] (R = Me [1], Et (**1**), *^i^*Pr (**2**)) the essentially equimolar ratio of homochiral ((*R**,*R**)-[R_2_Ga(*µ*-OCH(Me)CO_2_Me)]_2_) to heterochiral ((*R*,*S*)-[R_2_Ga(*µ*-OCH(Me)CO_2_Me)]_2_) species was observed irrespective of the alkyl group, the alkyl groups considerably affected the distribution of homo- and heterochiral species in the presence of pyridine derivatives. Therefore, gallium-bonded alkyl groups can be considered responsible for the distribution of homo- and heterochiral species, as well as for the pyridine derivatives of different coordination efficiencies to gallium.

### 2.2. Structure of (*S*,*S*)-[R_2_Ga(µ-OCH(Me)CO_2_Me]_2_ (R = Et((*S*,*S*)-***1***), ^i^Pr((*S*,*S*)-***2***))

(*S*,*S*)-[Et_2_Ga(*µ*-OCH(Me)CO_2_Me]_2_ ((*S*,*S*)-**1**) (Figure 4) and (*S*,*S*)-[*^i^*Pr_2_Ga(*µ*-OCH(Me)CO_2_(PLA)]_2_ ((*S*,*S*)-**2**) (Figure 5) crystallized as dimers, with the coordination sphere of both gallium atoms adopting a distorted trigonal-bipyramidal geometry, similarly to the previously reported (*S*,*S*)-[Me_2_Ga(*µ*-OCH(Me)CO_2_Me)]_2_ complex [42]. For a whole series of compounds—(*S*,*S*)-[R_2_Ga(*µ*-OCH(Me)CO_2_Me)]_2_ (R = Me, Et, *^i^*Pr)—the carbons of alkyl groups and alkoxide oxygen of the *melac* ligand defined the equatorial plane, and the carbonyl oxygen of the *melac* ester’s functionality and the bridging alkoxide oxygen of the second monomeric unit were located in the axial positions. Notably, the Ga–C bond distances of (*S*,*S*)-[R_2_Ga(*µ*-OCH(Me)CO_2_Me)] decreased in series *^i^*Pr ((*S*,*S*)-**2**; 1.985–1.996 Å) > Et ((*S*,*S*)-**1**; 1.962–1.978 Å) > Me (1.952–1.962 Å). Similarly, the Ga–O bridging bonds within the Ga_2_O_2_ central ring were significantly longer in the case of (*S*,*S*)-**2** (2.055(3) and 2.053(3)) and (*S*,*S*)-**1** (2.021(4) and 2.058(3)), in comparison with (*S*,*S*)-[Me_2_Ga(*µ*-OCH(Me)CO_2_Me)] (2.0269(18) and 2.0349(17) Å). Both findings were in line with the differences between the structures of **2** and (*R*,*S*)-[Me_2_Ga(*µ*-OCH(Me)CO_2_Me)] [1] discussed above. Importantly, for both (*S*,*S*)-**1** and (*S*,*S*)-**2**, the X-ray diffraction analyses revealed the presence of asymmetric dimers, which is in line with the structure of the previously characterized (*S*,*S*)-[Me_2_Ga(*µ*-OCH(Me)CO_2_Me)]_2_ complex [1,42]. Similarly to the latter, the observed asymmetry was the effect of the differently distorted methine carbon of *melac* ligands from the GaOCCO planes (Figure 6), which could be quantified by measuring the Ga–O–Ga–CH(Me) torsion angles [1]. However, the difference between the two different torsion angles Ga–O–Ga–CH(Me) in dimeric (*S*,*S*)-**1** (Figure 6b) and (*S*,*S*)-**2** (Figure 6c) were much smaller in comparison with (*S*,*S*)-[Me_2_Ga(*µ*-OCH(Me)CO_2_Me]_2_ (Figure 6a). Therefore, the presence of more sterically hindered Et and *^i^*Pr, in comparison with Me groups, resulted in significantly smaller distortion from the symmetric species, for which no or an equal level of distortion of both methine carbon *S-melac* ligands from the GaOCCO planes was expected. It is impossible to indicate unequivocally all the factors responsible for the above observation at this point in our research. However, one should note the presence of the center of symmetry in (*S*,*S*)-**1** and (*S*,*S*)-**2** resulting from the arrangement of ethyl and *iso*-propyl groups, respectively, in cases where the *S-melac* ligands are not taken under consideration (Figure 5). Such an effect was not evidenced for small gallium-bonded Me groups in (*S*,*S*)-[Me_2_Ga(*µ*-OCH(Me)CO_2_Me]_2_. Therefore, in the case of (*S*,*S*)-[R_2_Ga(*µ*-OCH(Me)CO_2_Me]_2_ (R = Me, Et, *^i^*Pr) complexes, the alkyl groups resulted in the formation of species of different symmetry, although the *C*_2_ point group was initially expected due to the presence of *S-melac* ligands of the same absolute configuration. Notably, the presence of two different structural elements, i.e., the alkyl groups on the gallium and *S-melac* ligands, which favored the formation of (*S*,*S*)-[R_2_Ga(*µ*-OCH(Me)CO_2_Me]_2_ (R = Et, *^i^*Pr) complexes of different symmetry, could be responsible for the higher tension within their central Ga_2_O_2_ ring in comparison with the analogous dimethyl derivative. This was reflected by the Ga(O1)–O(1)–Ga(1B)–O(1B) torsion angles of 3.3° ((*S*,*S*)-2) and 2.4° ((*S*,*S*)-1), larger in comparison with 2.0° in the case of (*S*,*S*)-[Me_2_Ga(*µ*-OCH(Me)CO_2_Me]_2_. The increased preference for the formation of centrosymmetric species for more bulky alkyl groups, should also be expected for the heterochiral (*R*,*S*)-[R_2_Ga(*µ*-OCH(Me)CO_2_Me]_2_ (R = Me, Et, *^i^*Pr) complexes discussed above, for which the presence of *melac* ligands of the opposite absolute configuration does not prevent the formation of centrosymmetric complexes. Such reasoning is in line, e.g., with the lower tendency for the formation of homochiral (*R**,*R**)-[R_2_Ga(*µ*-OCH(Me)CO_2_Me]_2_ (R = Me, Et, *^i^*Pr) in solution, presented and discussed above in the case of *rac*-[R_2_Ga(*µ*-OCH(Me)CO_2_Me]_2_, **1** and **2**.

In solution, the ^1^H NMR spectrum of (*S*,*S*)-**1** revealed one set of signals, which was in line with the presence of (*S*,*S*)-[Et_2_Ga(*µ*-OCH(Me)CO_2_Me]_2_. Additionally the dimeric structure of **1**, both in the solid state and solution, supported the proposed structure of (*S*,*S*)-**1**, which was additionally in agreement with the tendency of dialkylgallium alkoxides to form dimeric species [39]. Similarly, the major set of signals evidenced by the ^1^H NMR spectrum of (*S*,*S*)-**2**, was assigned to (*S*,*S*)-[*^i^*Pr_2_Ga(*µ*-OCH(Me)CO_2_Me]_2_. In this case, the additional minor septet corresponding to methine protons of the GaCH(CH_3_)_2_ groups (Appendix A) could be assigned to monomeric species, tentatively proposed for **2**. It can also indicate the presence of an equilibrium between dimeric (*S*,*S*)-[*^i^*Pr_2_Ga(*µ*-OCH(Me)CO_2_Me]_2_ and monomeric [*^i^*Pr_2_Ga((*S)*-OCH(Me)CO_2_Me] species, shifted considerably towards the former species. While the structure of (*S*,*S*)-**1** and (*S*,*S*)-**2** in solution, revealed by NMR spectra, was in line with the structure of *rac*-**1** and *rac*-**2**, discussed above, infrared spectroscopy was especially indicative of the effect of alkyl groups on the chelate Ga…O=C interaction of (*S*,*S*)-[R_2_Ga(*µ*-OCH(Me)CO_2_Me)]_2_ (R = Me, Et, *^i^*Pr). The FTIR of both (*S*,*S*)-**1** and (*S*,*S*)-**2** in CH_2_Cl_2_ revealed the presence of absorption bands of C=O groups at 1724 cm^−1^, essentially the same, accounting for experimental error, in comparison with (*S*,*S*)-[Me_2_Ga(*µ*-OCH(Me)CO_2_Me)]_2_ (ν_C=O_ = 1725 cm^−1^) [42]. Moreover, the deconvolution of the C=O band, leading to bands at 1724 and 1740 cm^−1^, corresponding to the coordinated and uncoordinated C=O groups of the *melac* ligand, was indicative of the percentage of the latter. In the case of (*S*,*S*)-**1**, it was revealed that 88% of C=O groups remained coordinated to gallium, which was exactly the same value as in the case of (*S*,*S*)-[Me_2_Ga(*µ*-OCH(Me)CO_2_Me)]_2_ [42]. A slightly higher percentage of coordinated C=O groups was observed for (*S*,*S*)-**2** (90%), which could explain the slightly stronger chelate Ga–O bonds in the case of **2** in comparison with both **1** and [Me_2_Ga(*µ*-OCH(Me)CO_2_Me)]_2_, as revealed by X-ray analysis.

### 2.3. Structure of rac-[R_2_Ga(µ-OCH(Me)C_5_H_4_N]_2_ (R = Et (***3***), ^i^Pr (***4***))

Complexes **3** and **4** were characterized in the solid state by X-ray diffraction analysis, which revealed, in both cases, the presence of centrosymmetric heterochiral dimers (*R*,*S*)-[R_2_Ga(*µ*-OCH(Me)C_5_H_4_N]_2_ (Figure 7 and Figure 8). This was in sharp contrast to the analogous dimethyl derivatives, which led upon crystallization solely to the (*R**,*R**)-[Me_2_Ga(*µ*-OCH(Me)C_5_H_4_N]_2_ of the *C*_2_ symmetry [1]. However, the observed structure of **3** and **4** was in line with the effect of the ethyl and *iso*-propyl groups on the tendency for the formation of centrosymmetric dialkylgallium alkoxide species, as shown above for the [R_2_Ga(*µ*-OCH(Me)CO_2_Me]_2_ complexes. In the case of **3** and **4**, five-coordinate gallium atoms adopted a distorted trigonal-bipyramidal geometry. Carbon atoms of gallium-bonded alkyl groups and the alkoxide oxygen of the OCH(Me)C_5_H_4_N ligand defined the equatorial plane. The nitrogen of the pyridine functionality of the OCH(Me)C_5_H_4_N ligand and a bridging alkoxide oxygen atom of the second monomeric unit were located in the axial positions. Notably, the presence of ethyl, and especially *iso*-propyl groups, did not essentially restrain the formation of Ga–N bonds to the fifth coordinate site of gallium in comparison to the analogous dimethyl derivative. This was demonstrated by the Ga–N distances for the following [R_2_Ga(*µ*-OCH_2_C_5_H_4_N)]_2_ complexes: 2.2789(18) Å (R = Me), 2.2908(11) Å (R = Et; **3**), and 2.2759(18) Å (R = *^i^*Pr; **4**). On the other hand, the presence of a *^i^*Pr group resulted in the elongation of the bridging Ga–O′ bond (2.1632(13) Å) within central Ga_2_O_2_ ring in comparison to the analogous bond in (*R**,*R**)-[Me_2_Ga(*µ*-OCH(Me)C_5_H_4_N)]_2_ (2.0931 Å), as well as in **3** (2.0906(8) Å).

In solution, the ^1^H NMR spectrum of **3** revealed one set of signals, which was not in agreement with either the molecular structure of (*R*,*S*)-**3** in the solid state or the mixture of (*R*,*S*)-**3** and (*R**,*R**)-**3** diastereomers, which was expected given the structures of **1** and **2** (see above). This could either suggest the presence of tentative monomeric—[Et_2_Ga(*µ*-OCH(Me)C_5_H_4_N)]_2_—or dimeric homochiral species—(*R**,*R**)-[Et_2_Ga(*µ*-OCH(Me)C_5_H_4_N)]_2_— in solution. The presence of the latter was further indicated by the ^1^H NMR spectrum of **4**, which revealed two sets of signals that, analogously to **2** and (*S*,*S*)-**2**, should be ascribed to the dimeric homochiral species—(*R**,*R**)-[*^i^*Pr_2_Ga(*µ*-OCH(Me)C_5_H_4_N)]_2_—and the tentative monomeric [*^i^*Pr_2_Ga(*µ*-OCH(Me)C_5_H_4_N)]_2_ species in a ratio of about 3:1. Although the ratio could not be determined precisely due to the overlapping signals (Appendix A), the presence of a higher percentage of monomeric species, in comparison with the structure of **2** and (*S*,*S*)-**2**, was in line with the structure of **2** in the presence of pyridine derivatives (see above), as well as with the interaction of pyridine functionality with Ga. Although the presence of only homochiral dimers—(*R**,*R**)-[R_2_Ga(*µ*-OCH(Me)C_5_H_4_N)]_2_ (R = Et, *^i^*Pr)—for **3** and **4** was initially surprising given the structure of **1** and **2**, this is in line with the suggested structure of *rac*-[Me_2_Ga(*µ*-OCH(Me)C_5_H_4_N)]_2_, as well as with the tendency of *rac*-[Me_2_Ga(*µ*-OCH(Me)CO_2_Me] for the formation of homochiral species in the presence of pyridines [1]. Despite the remaining questions and uncertainties concerning the structure of these complexes in solution, the minor effect of alkyl groups on the distribution of homo (*R**,*R**) and heterochiral (*R*,*S*) dimers for *rac*-[R_2_Ga(*µ*-OCH(Me)C_5_H_4_N)]_2_ (R = Me, Et, *^i^*Pr) was observed in contrast to their solid state structures. Notably, in the solid state, the tendency for the formation of centrosymmetric heterochiral dimers in the case of **3** and **4** was facilitated for diethyl- and especially di-*iso*-propylgallium *melac* derivatives in comparison to methyl derivatives [1]. This was in line with a reduced tendency for the formation of homochiral species with growing alkyl groups in the case of *rac*-[R_2_Ga(*µ*-OCH(Me)CO_2_Me)]_2_ (R = Me, Et (**1**) and *^i^*Pr (**2**)) in the presence of pyridine derivatives. The latter should be of special interest with regard to the ROP of *rac*-lactide using [R_2_Ga(*µ*-OCH(Me)CO_2_Me)]_2_, as well as the structure of the [R_2_Ga(*µ*-OCH(Me)CO_2_PLA)]_2_ propagating species.

### 2.4. Ring-Opening Polymerization of rac-LA with [R_2_Ga(µ-OCH(Me)CO_2_Me]_2_ (R = Me, Et, ^i^Pr)

In order to investigate the effect of the alkyl groups of [R_2_Ga(*µ*-OCH(Me)CO_2_Me]_2_ (R = Et, *^i^*Pr) on their activity, and especially their stereoselectivity in the ROP of *rac*-LA, we examined the catalytic properties of (*S*,*S*)-[R_2_Ga(*µ*-OCH(Me)CO_2_Me]_2_ (R = Et ((*S*,*S*)-**1**), *^i^*Pr ((*S*,*S*)-**2**)), chosen analogously to (*S*,*S*)-[Me_2_Ga(*µ*-OCH(Me)CO_2_Me]_2_, and previously used by us in polymerization studies [1,42]. Additionally, (*S*,*S*)-**1**/pyridine (1:6) and (*S*,*S*)-**2**/pyridine (1:6) were used in order to generate propagating species with the excess of homochiral—(*R**,*R**)—dimers and served as a representative examples of the catalytic systems—(*S*,*S*)-[R_2_Ga(*µ*-OCH(Me)CO_2_Me]_2_/*Py* (*Py*—pyridine derivative)—described above. One should note that after the insertion of *rac*-LA into the Ga–O bond of (*S*,*S*)-[R_2_Ga(*µ*-OCH(Me)CO_2_Me]_2_ (R = Et, *^i^*Pr), the formation of both heterochiral (*R*,*S*)-[R_2_Ga(*µ*-OCH(Me)CO_2_PLA]_2_ and homochiral (*R**,*R**)-[R_2_Ga(*µ*-OCH(Me)CO_2_PLA]_2_ propagating species should be expected, analogous to the previously described dimethylgallium derivatives (Figure 1) [1].

The ROP of *rac*-LA, with (*S*,*S*)-**1**, (*S*,*S*)-**1**/pyridine, (*S*,*S*)-**2**, (*S*,*S*)-**2**/pyridine, as well as (*S*,*S*)-[Me_2_Ga(*µ*-OCH(Me)CO_2_Me]_2_ and (*S*,*S*)-[Me_2_Ga(*µ*-OCH(Me)CO_2_Me]_2_/pyridine, was conducted at 40 °C and 70 °C. (Table 1) At 40 °C (Table 1, Entries 1–6), the insertion of *rac*-LA occurred solely into the Ga–O_alkoxide_ bond and led to the PLA with OH and *S*-*melac* end groups (Figure 4), which was in each case verified by ^1^H NMR and MALDI-TOF analysis (see the Appendix A). The latter indicated that there were essentially no transesterification reactions for the (*S*,*S*)-[R_2_Ga(*µ*-OCH(Me)CO_2_Me]_2_/pyridine catalytic systems, which confirmed the lack of activity of amines towards *rac*-LA at the polymerization conditions and the coordination insertion mechanism of the ROP of *rac*-LA with dialkylgallium alkoxide species. This was in line with previously reported results on the ROP of *rac*-LA with (*S*,*S*)-[Me_2_Ga(*µ*-OCH(Me)CO_2_Me]_2_ and (*S*,*S*)-[Me_2_Ga(*µ*-OCH(Me)CO_2_Me]_2_/pyridine [1]. Although, at 70 °C (Table 1, Entries 7–12), the insertion of *rac*-LA into Ga–O_alkoxide_ bond was evidenced both by the ^1^H NMR and MALDI TOF analysis, the higher tendency for side reactions, both the intermolecular transesterification and formation of cyclic PLA, were evidenced by the MALDI-TOF analyses. For a series of catalysts—(*S*,*S*)-[R_2_Ga(*µ*-OCH(Me)CO_2_Me]_2_ (R = Me, Et ((*S*,*S*)-**1**) and *^i^*Pr ((*S*,*S*)-**2**)—the side reactions were essentially eliminated in the case of (*S*,*S*)-**2**, most probably due to the presence of *^i^*Pr groups—the bulkiest among all investigated [R_2_Ga(*µ*-OCH(Me)CO_2_Me]_2_ (R = Me, Et, *^i^*Pr) complexes. For a series of catalytic systems—(*S*,*S*)-[Me_2_Ga(*µ*-OCH(Me)CO_2_Me]_2_/pyridine (1:6), (*S*,*S*)-**1**/pyridine (1:6) and (*S*,*S*)-**2**/pyridine (1:6)—the side reactions were more pronounced; however, the lowest extent was observed in the case of di-*iso*-propyl species. Although the reader must note that the MALDI-TOF analysis is not a quantitative method, the effect of *^i^*Pr groups on the decreased tendency for side reactions, especially transesterification, should be considered. The differences in the activities of both (*S*,*S*)-[R_2_Ga(*µ*-OCH(Me)CO_2_Me]_2_ catalysts, at 70 °C, and (*S*,*S*)-[R_2_Ga(*µ*-OCH(Me)CO_2_Me]_2_/pyridine (1:6) catalytic systems, both at 40 °C and 70 °C, were too small to substantially demonstrate the effect of the alkyl groups. However, in the case of the (*S*,*S*)-[R_2_Ga(*µ*-OCH(Me)CO_2_Me]_2_ catalysts, at 40 °C, and in the absence of pyridine (Table 1, Entries 1–3), for which the lowest activity of propagating species was observed, the *rac*-LA conversions decreased in the following series: (*S*,*S*)-**2** > (*S*,*S*)-**1** > (*S*,*S*)-[Me_2_Ga(*µ*-OCH(Me)CO_2_Me]_2_. Such an observation could be the result of the decreasing distances of bridging Ga–O bonds with the decreasing bulkiness of the alkyl groups in series *^i^*Pr > Et > Me. While the highest activity was observed for (*S*,*S*)-**2**, we cannot exclude the effect of the presence of minor fractions of other species, tentatively ascribed as monomeric species, in this case.

For the ROP of *rac*-LA with [R_2_Ga(*µ*-OCH(Me)CO_2_Me]_2_ (R = Me, Et ((*S*,*S*)-**1**) and *^i^*Pr ((*S*,*S*)-**2**) (Table 1, Entries 1–3 and 7–12), only atactic PLA was formed, even at mild conditions at which no side reactions could be detected. However, in the case of catalytic systems [R_2_Ga(*µ*-OCH(Me)CO_2_Me]_2_ (R = Me, Et)/pyridine (1:6) (Table 1, Entries 4–6), the formation of heterotactically enriched PLA was observed. The essential lack of heteroselectivity at 70 °C (Table 1, Entries 10–12) should be considered a result of both high temperature as well as the effect of side reactions at polymerization conditions. Most importantly, the heteroselectivity, expressed as a probability of *racemo*-linkages (*P*_r_) in the formed PLA, strongly correlated with the excess of homochiral dimeric propagating species, which should be expected based on the structure studies of [R_2_Ga(*µ*-OCH(Me)CO_2_Me]_2_ (R = Me [1], Et ((*S*,*S*)-**1**) and *^i^*Pr ((*S*,*S*)-**2**) in the presence of pyridine (see the discussion above). Notably, essentially no stereoselectivity was observed in the case of (*S*,*S*)-**2** and (*S*,*S*)-**2**/pyridine, which indicates that there is essentially no effect of minor species, tentatively assigned to monomeric species, on the stereoselectivity. Most importantly, despite the limited number of examples, the obtained results clearly demonstrate that the chiral recognition induced the formation of homochiral dimeric (*R**,*R**)-[R_2_Ga(*µ*-OCH(Me)CO_2_PLA]_2_ propagating species and should be considered responsible for their stereoselectivity in the ROP of *rac*-LA. While the presence of bulky alkyl groups in [R_2_Ga(*µ*-OCH(Me)CO_2_PLA]_2_ has no adverse effect on their activity and can be somehow helpful in order to limit side reactions, it considerably decreases or even eliminates the excess of homochiral (*R**,*R**)-[R_2_Ga(*µ*-OCH(Me)CO_2_Me]_2_ species and therefore the stereoselectivity.

## 3. Materials and Methods

### 3.1. General Procedures

All operations were carried out under dry argon using standard Schlenk techniques. Solvents and reagents were purified and dried prior to use. Solvents were purified using MBRAUN Solvent Purification Systems (MB-SPS-800) and stored over molecular sieves. *rac*-Lactide was purchased from Sigma-Aldrich (Merck, Warsaw, Poland) and further purified by crystallization from anhydrous toluene and then sublimated. (*S*)-methyl lactate and (*rac*)-methyl lactate were purchased from Sigma-Aldrich. *rac*-HOCH(Me)C_5_H_4_N (*rac*-1-(2-pyridinyl)ethanol) was synthesized according to the literature [58]. All alcohols were stored under argon, over 4 Å molecular sieves, at least 48h before they were used for the reaction with trialkylgallium compounds. Me_3_Ga was purchased from STREM Chemicals, Inc. and used as received. Et_3_Ga and *^i^*Pr_3_Ga, which we received as a gift, were synthesized in the research group of Prof. Kazimierz B. Starowieyski at Warsaw University of Technology. *^i^*Pr_3_Ga was freshly distilled under vacuum prior to use. (*S*,*S*)-[Me_2_Ga(*μ*-OCH(Me)CO_2_Me)]_2_ was synthesized according to our article [42]. The ^1^H and ^13^C NMR spectra were recorded on Varian 700 MHz (Varian, Palo Alto, CA, USA) and Agilent 400 MHz spectrometers (Agilent Technologies, Santa Clara, CA, USA), with shifts given in ppm according to the deuterated solvent shift. FTIR spectra were recorded on a FTIR Perkin Elmer System 2000 instrument (Perkin Elmer, Waltham, MA, USA). MALDI-TOF spectra were recorded on a Bruker Model ultrafleXtreme instrument (Bruker, Billerica, MA, USA). Elemental analysis was performed on a Vario EL III instrument (Heraeus, Hanau, Germany) for (*S*,*S*)-**1** and (*S*,*S*)-**2**. Although, for (*S*,*S*)-**1**, the obtained values were low for carbon with 0.6%, they represent the best values we could obtain experimentally. EA for (*S*,*S*)-**2** was significantly low in carbon and hydrogen. This could result from the EA measurement or sample handling without an inert atmosphere.

### 3.2. Synthesis of Gallium Complexes

The representative method for the synthesis of **1**, (*S*,*S*)-**1**, **2**, (*S*,*S*)-**2**, **3**, and **4** was as follows. The stirred solution of trialkylgallium—R_3_Ga (R = Et, *^i^*Pr; 3.0 mmol)—in CH_2_Cl_2_ (1 mL) was added dropwise to the solution of *S*-methyl lactate/*rac*-methyl lactate/*rac*-1-(2-pyridinyl)ethanol (3.0 mmol) in CH_2_Cl_2_ (9 mL), at −40 °C. Subsequently, the cooling bath was removed, and the reaction mixture was warmed slowly to room temperature. Upon increasing the solution temperature, the evolution of gas was observed. The reaction mixture was stirred at room temperature for 5h. Then, the solvent and volatiles were removed under a vacuum to give, in each case, a white crystalline solid. (*S*,*S*)-**1** and (*S*,*S*)-**2** were recrystallized from n-hexane, and **1**, **2**, **3**, **4** from the methylene chloride/n-hexane solution, at −20 °C, to give colorless crystals in moderate to high yields, the latter depending on the concentration of the solution and CH_2_Cl_2_/n-hexane ratio.

(*S*,*S*)-**1**; ^1^H NMR (CD_2_Cl_2_, 700 MHz): 0.35, (q, ^3^*J*(H,H) = 8.0 Hz, 4H, GaCH_2_*CH*_3_), 1.04 (t, ^3^*J*(H,H) = 8.0 Hz, 6H, Ga*CH*_2_CH_3_), 1.39 (d, ^3^*J*(H,H) = 6.9 Hz, 3H, *CH*CH_3_), 3.77 (s, 6H, OCH_3_), 4.44 (q, ^3^*J*(H,H) = 6.9 Hz, 1H, CH*CH*_3_). ^13^C {^1^H} NMR (CD_2_Cl_2_, 175 MHz): 5.9, 10.2, 22.1, 53.3, 68.3, 179.1. FTIR: 1724 cm^−1^ (C=O). Anal. Calcd for C_16_H_34_Ga_2_O_6_: C, 41.61, H, 7.42. Found: C, 40.96, H, 7.32.

(*S*,*S*)-**2**; 0.83 (sept, ^3^*J*(H,H) = 7.6 Hz, 2H, GaCH(*CH*_3_)_2_), 1.12 (d, ^3^*J*(H,H) = 7.6 Hz, 12H, Ga*CH*(CH_3_)_2_), 1.42 (d, ^3^*J*(H,H) = 6.9 Hz, 3H, *CH*CH_3_), 3.78 (s, 3H, OCH_3_), 4.52 (q, ^3^*J*(H,H) = 6.9 Hz, 1H, CH*CH*_3_). ^13^C {^1^H} NMR (CD_2_Cl_2_, 175 MHz): 17.4, 21.7, 22.3, 53.5, 68.85, 179.69. FTIR: 1724 cm^−1^ (C=O).

**1**: ^1^H NMR (toluene-*d*_8_, 700 MHz): 0.61 (q, ^3^*J*(H,H) = 8.1 Hz, 1H, GaCH_2_*CH*_3_), 0.74–0.65 (m, 2H, GaCH_2_*CH*_3_), 0.77 (q, ^3^*J*(H,H) = 8.1 Hz, 1H, GaCH_2_*CH*_3_), 1.46–1.25 (m, 9H, *CH*CH_3_, Ga*CH*_2_CH_3_), 3.21, 3.22 (s, 3H, OCH_3_), 4.44–4.39 (m, 1H, CH*CH*_3_).

**2**: ^1^H NMR (toluene-*d*_8_, 700 MHz): 1.05–0.98 (sept, ^3^*J*(H,H) = 7.6 Hz, 1H, GaCH(*CH*_3_)_2_), 1.12–1.05 (sept, ^3^*J*(H,H) = 7.6 Hz, 2H, GaCH(*CH*_3_)_2_), 1.21–1.14 (m, 1H, GaCH(*CH*_3_)_2_), 1.50–1.33 (m, 30H, *CH*CH_3_, Ga*CH*(CH_3_)_2_), 3.22, 3.23 (s, 6H, OCH_3_), 4.52–4.47 (m, 2H, CH*CH*_3_).

**3**: ^1^H NMR (CD_2_Cl_2_, 400 MHz): 0.36 (q, ^3^*J*(H,H) = 8.0 Hz, 4H, GaCH_2_*CH*_3_), 0.96 (t, ^3^*J*(H,H) = 8.0 Hz, 6H, Ga*CH*_2_CH_3_), 1.52 (d, ^3^*J*(H,H) = 6.6 Hz, 3H, *CH*CH_3_), 5.23 (q, ^3^*J*(H,H) = 6.6 Hz, 1H, CH*CH*_3_), 7.37–7.29 (m, 2H, CH_(C5H4N)_), 7.84–7.77 (m, 1H, CH_(C5H4N)_), 8.43–8.37 (m, 1H, CH_(C5H4N)_).

**4**: ^1^H NMR (CD_2_Cl_2_, 400 MHz): 0.78–0.92 (m, 2H, GaCH(*CH*_3_)_2_, 1.06 (m, 12H, Ga*CH*(CH_3_)_2_), 1.49 (d, ^3^*J*(H,H) = 6.6 Hz, 3H, *CH*CH_3_), 5.33 (q, ^3^*J*(H,H) = 6.6 Hz, 1H, CH*CH*_3_), 7.33–7.49 (m, 2H, CH_(C5H4N)_), 7.88–7.95 (m, 1H, CH_(C5H4N)_), 8.27–8.33 (m, 1H, CH_(C5H4N)_).

### 3.3. General Procedure for the ROP of rac-LA with (*S*,*S*)-***1*** and (*S*,*S*)-***2***

To a *rac*-LA (334.8 mg, 2.32 mmol, 25 eq) solution in CH_2_Cl_2_, or a suspension in toluene, (10 mL) the solution of the catalyst (0.09 mmol, 1 eq Ga) in the same solvent was added. The reaction was stirred at the indicated temperature for 96 h in order to support high conversions. For the polymerizations in toluene, the dissolution of rac-LA was observed with its conversion, while after 96 h, the whole rac-LA was dissolved. Each polymerization was quenched by the addition of a HCl solution (5%, 50 mL). The organic phase was separated, washed twice with water (50 mL), dried over anhydrous MgSO_4_, and dried under vacuum to give PLA as a white solid. ^1^H NMR (CDCl_3_, 400 MHz): (a) PLA signals, 1.52–1.60 (m, 3H, *CH*CH_3_), 5.11–5.26 (m, 1H, CH*CH*_3_); (b) end groups: 1.47–150 (m, *CH*CH_3_), 3.73, 3.74 (s, OCH_3_), 4.31–4.38 (m, CH*CH*_3_).

### 3.4. Crystal Structure Determination

Select crystals of **1**–**3** were measured on the Bruker D8 Venture single crystal diffractometer equipped with a CMOS II detector and LT device using a Mo sealed tube as a radiation source (*λ* = 0.71073 Å). The samples were measured at lowered temperatures as follows: 100 K for (*R*,*S*)-**3** and 130 K for (*S*,*S*)-**1**, (*R*,*S*)-**2**, and (*S*,*S*)-**2**. The data were collected using Bruker APEX3 software [59], followed by the integration in the Bruker SAINT program [60] and scaled in Bruker SADABS [61], [samples: (*R*,*S*)-**3**, (*S*,*S*)-**1**, (*R*,*S*)-**2**] or TWINABS [62], [sample: (*S*,*S*)-**2**]. A single crystal of (*R*,*S*)-**4** was measured on an Oxford Diffraction κ-CCD Gemini A Ultra diffractometer. Cell refinement and data collection, as well as data reduction and analysis, were performed with Rigaku Oxford CRYSALIS^PRO^ software (Rigaku Oxford Diffraction CrysAlisPro Software System, 2024, v. 1.171.43.136a). Using Olex2 [63], the structure was solved with ShelXT [64], structure solution program using Intrinsic Phasing, and refined with ShelXL [65], a refinement package using least squares minimization. All non-hydrogen atoms were refined with anisotropic displacement parameters. Hydrogen atoms attached to carbon atoms were added to the structure model at geometrically idealized coordinates and refined as riding atoms. Flack parameters were calculated from selected quotients (Parsons’ method) [66]. CCDC2406865-2406869 contains the supplementary crystallographic data for this paper. These data can be obtained free of charge via https://www.ccdc.cam.ac.uk/structures/? (accessed on 1 January 2025) (or from the CCDC, 12 Union Road, Cambridge CB2 1EZ, UK; Fax: +44 1223 336033; E-mail: deposit@ccdc.cam.ac.uk).

Crystal Data for (*R*,*S*)-**2** C_20_H_42_Ga_2_O_6_ (*M* = 517.97 g/mol): monoclinic, space group P2_1_/c (no. 14), *a* = 10.5816(5) Å, *b* = 15.3600(5) Å, *c* = 7.9356(3) Å, *β* = 100.808(2)°, *V* = 1266.92(9) Å^3^, *Z* = 2, *T* = 130.00 K, μ(MoKα) = 2.155 mm^−1^, *Dcalc* = 1.358 g/cm^3^, 37,190 reflections measured (5.86° ≤ 2Θ ≤ 56.5°), 3118 unique (*R*_int_ = 0.0291, R_sigma_ = 0.0133), used in all calculations. The final *R*_1_ was 0.0208 (I > 2σ(I)) and *wR*_2_ was 0.0485 (all data).

Crystal Data for (*S*,*S*)-**2** C_20_H_42_Ga_2_O_6_ (*M* = 517.97 g/mol): monoclinic, space group P2_1_ (no. 4), *a* = 7.8664(2) Å, *b* = 15.5512(4) Å, *c* = 10.4906(3) Å, *β* = 100.1397(9)°, *V* = 1263.29(6) Å^3^, *Z* = 2, *T* = 130.0(5) K, μ(MoKα) = 2.161 mm^−1^, *Dcalc* = 1.362 g/cm^3^, 33,428 reflections measured (5.88° ≤ 2Θ ≤ 50.76°), 4573 unique (*R*_int_ = 0.0203, R_sigma_ = 0.0065), used in all calculations. The final *R*_1_ was 0.0209 (I > 2σ(I)) and *wR*_2_ was 0.0668 (all data).

Crystal Data for (*S*,*S*)-**1** C_16_H_34_Ga_2_O_6_ (*M* = 461.87 g/mol): triclinic, space group P1 (no. 1), *a* = 8.1834(7) Å, *b* = 8.2430(7) Å, *c* = 8.7070(8) Å, *α* = 113.352(3)°, *β* = 90.424(3)°, *γ* = 94.829(3)°, *V* = 536.78(8) Å^3^, *Z* = 1, *T* = 130.0(5) K, μ(MoKα) = 2.534 mm^−1^, *Dcalc* = 1.429 g/cm^3^, 15,512 reflections measured (5.76° ≤ 2Θ ≤ 52.98°), 4348 unique (*R*_int_ = 0.0313, R_sigma_ = 0.0330), used in all calculations. The final *R*_1_ was 0.0186 (I > 2σ(I)) and *wR*_2_ was 0.0466 (all data).

Crystal Data for (*R*,*S*)-**3** C_22_H_36_Ga_2_N_2_O_2_ (*M* = 499.97 g/mol): triclinic, space group *P* $\bar{1}$ (no. 2), *a* = 8.7984(5) Å, *b* = 8.8724(5) Å, *c* = 8.9171(5) Å, *α* = 82.408(2)°, *β* = 66.358(2)°, *γ* = 67.060(2)°, *V* = 587.00(6) Å^3^, *Z* = 1, *T* = 100.00 K, μ(MoKα) = 2.313 mm^−1^, *Dcalc* = 1.414 g/cm^3^, 32,854 reflections measured (4.988° ≤ 2Θ ≤ 59.996°), 3435 unique (*R*_int_ = 0.0215, R_sigma_ = 0.0108), used in all calculations. The final *R*_1_ was 0.0155 (I > 2σ(I)) and *wR*_2_ was 0.0421 (all data).

Crystal Data for (*R*,*S*)-**4** C_26_H_44_Ga_2_N_2_O_2_ (*M* = 556.07 g/mol): monoclinic, space group P2_1_/n (no. 14), *a* = 10.0033(3) Å, *b* = 12.2497(4) Å, *c* = 11.8020(4) Å, *β* = 94.142(3)°, *V* = 1442.41(8) Å^3^, *Z* = 2, *T* = 293.15 K, μ(Mo Kα) = 1.890 mm^−1^, *Dcalc* = 1.280 g/cm^3^, 20,992 reflections measured (6.652° ≤ 2Θ ≤ 55.752°), 3440 unique (*R*_int_ = 0.0283, R_sigma_ = 0.0175), used in all calculations. The final *R*_1_ was 0.0295 (I > 2σ(I)) and *wR*_2_ was 0.0827 (all data).

## 4. Conclusions

In summary, we have focused on the effect of alkyl substituents on the structure of *rac*-[R_2_Ga(*µ*-OCH(Me)CO_2_Me)]_2_, (*S*,*S*)-[R_2_Ga(*µ*-OCH(Me)CO_2_Me)]_2_, and *rac*-[R_2_Ga(*μ*-OCH(Me)C_5_H_4_N)]_2_ (R = Me, Et, *^i^*Pr), as well as the catalytic properties of (*S*,*S*)-[R_2_Ga(*µ*-OCH(Me)CO_2_Me)]_2_ and (*S*,*S*)-[R_2_Ga(*µ*-OCH(Me)CO_2_Me)]_2_/pyridine in the ROP of *rac*-LA. We expected these studies to be especially indicative for the effect of alkyl substituents of [R_2_Ga(*µ*-OCH(Me)CO_2_PLA)]_2_ (PLA—growing polylactide chain) propagating species on their tendency to form homochiral species and the influence of the latter on their stereoselectivity in the ROP of *rac*-LA.

In solution, *rac*-[R_2_Ga(*µ*-OCH(Me)CO_2_Me)]_2_ (R = Et, *^i^*Pr) have shown a much lower tendency for the formation of homochiral (*R**,*R**)-[R_2_Ga(*µ*-OCH(Me)CO_2_Me)]_2_ species in the presence of pyridine in comparison with the previously reported *rac*-[Me_2_Ga(*µ*-OCH(Me)CO_2_Me)]_2_ complexes [1]. The analogue’s effect has been demonstrated by the formation of heterochiral *R*,*S*-[R_2_Ga(μ-OCH(Me)C_5_H_4_N)]_2_, (R = Et, *^i^*Pr), in contrast to homochiral (*R**,*R**)-[Me_2_Ga(μ-OCH(Me)C_5_H_4_N)]_2_ [1], upon crystallization_._ Notably, the X-ray analysis of the homochiral (*S*,*S)*-[R_2_Ga(*µ*-OCH(Me)CO_2_Me)]_2_ (R = Me, Et, *^i^*Pr) complexes has further proven the greater tendency of diethyl and di-*iso*-propyl derivatives for the formation of centrosymmetric species, which affects their structures considerably. Importantly, the decreasing tendency to form homochiral species (*R**,*R**)-[R_2_Ga(*µ*-OCH(Me)CO_2_Me)]_2_ with larger alkyl substituents on gallium was in line with the hetereoselectivity of the (*S*,*S*)-[R_2_Ga(*µ*-OCH(Me)CO_2_Me)]_2_ and (*S*,*S*)-[R_2_Ga(*µ*-OCH(Me)CO_2_Me)]_2_ (R = Me, Et, *^i^*Pr)/pyridine catalytic systems in the ROP of *rac*-LA, which decreased in series R = Me > Et > *^i^*Pr. The observed correlation supports the crucial role of homochiral (*R**,*R**)-[R_2_Ga(*µ*-OCH(Me)CO_2_R)]_2_ species on the heteroselectivity of dialkylgallium alkoxides, as previously reported by us [1]. Notably, the findings described in this article constitute a new example indicating that chiral recognition induced the formation of homochiral dimeric metal alkoxides, which can be used in order to tune the stereoselectivity in the coordination polymerization of lactide and other chiral heterocyclic monomers that interest us.

## Data Availability

The data supporting this article have been included as part of the manuscript and in the Appendix A.

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
