# Peer review of "The Effect of Alkyl Substituents on the Formation and Structure of Homochiral (R*,R*)-[R2Ga(µ-OCH(Me)CO2R′)]2 Species—Towards the Factors Controlling the Stereoselectivity of Dialkylgallium Alkoxides in the Ring-Opening Polymerization of rac-Lactide"

_molecules, 2025, doi:10.3390/molecules30010190_

Round 1
Reviewer 1 Report
Comments and Suggestions for Authors
Horeglad et al. report on the role of alkyl substituents in influencing the structure, chiral recognition, and catalytic activity of dialkylgallium alkoxides in the ring-opening polymerization (ROP) of racemic lactide (rac-LA). It is highly interesting to see that bulkier alkyl groups reduce the formation of homochiral complexes, resulting in lower stereoselectivity. Hence, homochiral dimers are essential for achieving heteroselectivity in ROP. Moreover, bulkier alkyl substituents improve resistance to side reactions, such as transesterification.
The study uses standard techniques like X-ray diffraction, NMR, and FTIR to offer detailed structural and mechanistic insights. It reveals the critical role of chiral recognition and alkyl group bulkiness in catalyst performance.
The findings can guide the development of more effective catalysts for producing biodegradable polymers with tailored properties.
The study would be way stronger when kinetics are included. In the present state, the influence of chirality on the polymer is revealed but normally (as shown by many groups, Kol, Tolman etc.) the kinetics are also heavily influenced. This should be studied and included into the manuscript.
Afterwards, the manuscript can be accepted for publication.
Author Response
Dear Reviewer. Thank you for the appreciation of our work and a valuable suggestion concerning the kinetic studies. We agree that kinetics would always constitute an interesting piece of data in the structure-activity studies of metal complexes in ROP; in our case dialkylgallium complexes. However, we would like to stress that in this case we have focused on the structure of gallium complexes and the relationship between the latter and their stereoselectivity. We also regard this issue a key point of our article. Therefore, we believe that although kinetic studies could be interesting, it does not remain the main issue here. Our reasoning seems true in the light of your suggestion that this part of our article only “can be improved”. Therefore, we would like not to include kinetic studies in this manuscript. We are currently working on the article, in which we are going to sum up our results on the catalytic activity of dialkylgallium alkoxides in the ROP and especially immortal ROP of lactide. We are sure that such an article would be an ideal place for suggested kinetic studies, notably a laborious piece of work.
Reviewer 2 Report
Comments and Suggestions for Authors
I find this to be an excellent paper, investigating in depth the structure and reactivity of gallium complexes of relevance to lactic acid polymerisation. Important evidence has been obtained both from detailed NMR studies and X-ray structure determination and these are thoroughly analysed and discussed. The paper is well written and appropriately referenced. I recommend acceptance subject only to a few minor corrections and changes to improve the wording as follows:
Line 124 - change 'Except of' to 'Except for'
Line 126 - change 'concerned' to 'aware'
Line 249 - change 'irrespectively to' to irrespective of the'
Line 252 - change 'equally to' to 'equally for'
Line 524 - change 'molecular sieves 4Å' to '4Å molecular sieves'
Line 536 - change 'EA for (S,S)' to 'EA for (S,S)-2'
Line 767 - remove 'Error! Bookmark not defined'
SI
Caption of Figure S11 - change '(S,S)-1' to '(S,S)-2'
X-ray data, Table S1 - entries have jumped the line under unit cell dimensions in last 2 columns
Space group for (R,S)-4 change P21/2 to P21/n
Author Response
Dear Reviewer. Thank you for the appreciation of our work and remarks. We have corrected the manuscript accordingly.
Round 2
Reviewer 1 Report
Comments and Suggestions for Authors
The authors state that the kinetics will be part of an upcoming publication where they are studying the immortal type or ROP. I fully understand the point now. Immortal ROP is very important and a lot of kinetics are needed. Hence, this would be too much for the present manuscript. Thus, the manuscript can be accepted.